# Airborne Transmission of Foot-and-Mouth Disease Virus: A Review of Past and Present Perspectives

**DOI:** 10.3390/v14051009

**Published:** 2022-05-09

**Authors:** Emma Brown, Noel Nelson, Simon Gubbins, Claire Colenutt

**Affiliations:** 1The Pirbright Institute, Ash Road, Pirbright, Surrey GU24 0NF, UK; simon.gubbins@pirbright.ac.uk (S.G.); claire.colenutt@pirbright.ac.uk (C.C.); 2The Met Office, FitzRoy Road, Exeter, Devon EX1 3PB, UK; noel.nelson@metoffice.gov.uk

**Keywords:** FMDV, aerosols, airborne, transmission

## Abstract

The primary transmission route for foot-and-mouth disease (FMD), a contagious viral disease of cloven-hoofed animals, is by direct contact with infected animals. Yet indirect methods of transmission, such as via the airborne route, have been shown to play an important role in the spread of the disease. Airborne transmission of FMD is referred to as a low probability- high consequence event as a specific set of factors need to coincide to facilitate airborne spread. When conditions are favourable, airborne virus may spread rapidly and cause disease beyond the imposed quarantine zones, thus complicating control measures. Therefore, it is important to understand the nature of foot-and-mouth disease virus (FMDV) within aerosols; how aerosols are generated, viral load, how far aerosols could travel and survive under different conditions. Various studies have investigated emissions from infected animals under laboratory conditions, while others have incorporated experimental data in mathematical models to predict and trace outbreaks of FMD. However, much of the existing literature focussing on FMDV in aerosols describe work which was undertaken over 40 years ago. The aim of this review is to revisit existing knowledge and investigate how modern instrumentation and modelling approaches can improve our understanding of airborne transmission of FMD.

## 1. Foot-and-Mouth Disease Virus Transmission

Foot-and-mouth disease virus (FMDV) (family *Picornaviridae*, genus *Aphthovirus*) causes a highly infectious and contagious disease of wild and domesticated cloven-hoofed animals, with outbreaks causing considerable economic consequences for the livestock industry worldwide. The disease is endemic in parts of Africa, Asia, the Middle East and South America, and sporadically causes outbreaks in previously free countries and regions [1]. Outbreaks of FMD can be devastating to the agricultural industry due to the expense of implementing control measures and the consequent trade restrictions which prevent the export of animals and animal products [2]. It is estimated that FMD costs between US$6.5–21 billion per year in endemic regions [3]. The impact of FMD outbreaks on the farming industry in FMD-free regions was illustrated during the UK epidemic in 2001. Over an 8-month period, approximately 6.5 million animals were slaughtered, and the epidemic was estimated to have cost the UK economy in excess of £8 billion in agricultural losses and restrictions on tourism [2,4].

The main transmission route for the disease is by inhalation of virus particles through direct contact with the breath of acutely infected animals [5]. Transmission can also occur indirectly via a contaminated environment where FMDV can survive for prolonged periods of time under favourable conditions [6,7]. Ideal conditions for virus survival are temperatures below 50 °C, relative humidity above 55% and neutral pH [6,7,8]. Airborne transmission has also been implicated in the spread of disease over both long (considered to be up to 50 km over land and 200 km over water) and short distances (within premises and neighbouring premises within 2 km proximity of each other) [9,10,11,12]. Airborne transmission has been widely studied for FMDV, but there are still large gaps in our knowledge regarding the practical applications for FMD control and how the use of modern instrumentation and modelling can aid our understanding of airborne transmission of FMD. The aim of this review is to revisit existing knowledge and identify gaps which could direct future studies in this area.

## 2. Definition of Aerosol vs. Droplet Transmission

When studying bioaerosols generated by human or animal hosts it is important to distinguish between droplet and airborne transmission of an infectious agent. It is generally accepted that droplet transmission is a form of direct contact transmission in which respiratory droplets measuring > 5 μm in diameter travel directly from the respiratory tract of an infectious host to a susceptible host, over short distances (<1 m) [13,14]. By contrast, airborne transmission is defined as respirable particles which are exhaled from an infectious host and partially evaporate in the surrounding air [15]. These small, partially evaporated particles measuring < 5 μm in diameter can remain infectious over time and can be dispersed over long distances (>1 m) by air currents, potentially causing long distance transmission events [13,14]. However, the authors acknowledge that in light of research carried out during the COVID-19 pandemic the current definitions for aerosol and droplet transmission have been contested and new definitions have been proposed. Under the new definition, aerosol transmission is used to describe particles which are less than 100 µm in diameter and droplet transmission is defined as particles over 100 µm in diameter [16,17]. For the purpose of this review we will focus on airborne transmission of FMDV, which given the potential distances involved in the airborne spread of the virus, we define as particles measuring < 5 μm in diameter. Droplet transmission will not be discussed as we define this as a direct contact route which does not fall within the scope of this review.

## 3. Airborne Transmission of FMD and Major Outbreaks

Emitted infectious aerosols can be spread via atmospheric transport causing long distance transmission events which pose a serious threat to the control of FMD outbreaks [9,10,11]. Airborne virus can be spread rapidly causing disease beyond the imposed quarantine zones and, as such, transmission via this route can be considered a low probability-high consequence event [18]. Various factors are needed for the transmission of virus over long distances (Figure 1) such as (i) high virus emission, most likely from pigs in the acute stages of disease, (ii) prevailing weather conditions conducive to low virus aerosol dispersion, such as gentle winds and a stable atmosphere that will suppress turbulent mixing, and therefore eliminate the upward motion of the virus aerosol, (iii) high virus survival such as a relative humidity of 55% or more (iv) large numbers of susceptible livestock exposed to a virus plume for many hours, often cattle due to their low infectivity threshold [11].

Airborne transmission is generally assigned as the route of transmission where no other route is thought possible, for example if there are no animal movements from infected farms or shared equipment or personnel. Therefore, meteorological and epidemiological investigations are undertaken to explore the likelihood of airborne transmission, for example do the general wind directions match the sequence of disease spread among the infected premises. These investigations have indicated that airborne transmission was the most probable cause of several historic FMD outbreaks. For example, FMD infections in cattle occurring on farms in Jersey in 1974 and 1981 and the Isle of Wight in 1981, are thought to have originated from outbreaks in Brittany, France [19]. Epidemiological evidence collected at that time concluded that airborne transmission was the only possible route of infection. During these outbreaks virus would have been transported over distances of 500 and 300 km respectively [19]. Airborne transmission of FMD also occurred during the 1967–1968 epidemic in the UK, in which airborne dispersion over land reached approximate distances ranging from 60 to 150 km [12]. Airborne transmission most notably occurred during the UK epidemic of FMD in 2001, when virus carrying aerosols were transmitted between farms in close vicinity of each other causing infection in susceptible livestock [10].

## 4. Existing Knowledge from Literature: What We Know and How We Know It

### 4.1. Susceptibility to Airborne Infection

The difference between species in susceptibility and the quantity of virus emitted is well documented in the available literature. Previous studies have shown as little as 10 TCID_50_ is required to initiate an infection in susceptible ruminants [20]. Cattle are considered most susceptible to FMDV infection via the airborne route as their inhaled dose is likely to be larger than other livestock species due to their greater lung capacity [21,22]. By contrast, swine require a much higher dose than ruminants and it has been estimated that 6000 TCID_50_ are required to cause an infection [23].

Generally, research which aims to assess the susceptibility of different species to airborne virus describe an indirect transmission experimental design. This consists of placing a needle-inoculated animal in a cabinet and exposing naïve animals to the excreted aerosols via a fitted mask which is connected to the cabinet through an exposure tunnel [23,24,25]. This type of experimental design allows the naïve animals to be exposed to aerosols generated from infected animals which simulates the natural route of infection and means results obtained from the studies are more representative of real-life scenarios.

Studies which aim to measure the efficiency of aerosols as a route of infection also describe an experimental set up designed to mimic a natural route of infection (Figure 2). These studies have shown infectious FMDV and FMDV RNA was detected from the oral cavities of cattle [26] and sheep [27,28] after they were challenged using nebulised virus and a fitted face mask, with delivery of virus aerosols into the nostrils of recipient animals. Additionally, between-pen transmission experiments have been performed whereby needle-inoculated donor animals are placed in one room and contact animals in adjoining pens with a free flow of air between the pens. Results from these types of experiments have shown transmission was possible from pigs to cattle [29] and pigs to pigs [30,31] but the rate of transmission was reduced compared to within-pen transmission, which is likely to occur via direct transmission.

### 4.2. Virus Emissions from Infected Animals

Although swine have been shown to be less susceptible to airborne FMDV infection, infected swine are an important source of aerosolised FMDV as they are capable of excreting 100 to 1000 times more virus than infected sheep or cattle [19,22,23]. In the grand scheme of things very few strains of FMDV have been used in studies that aim to quantify emissions from FMD–infected animals. However, where experimental work has been carried out emissions from pigs were higher than for other susceptible species and the C Noville strain of FMDV was emitted in the highest titres compared to other strains regardless of the species involved. The highest emissions recorded for C Noville were 10^8.6^ TCID_50_ and 10^7.6^ TCID_50_ in pigs [21,32]. By comparison, serotype O strains are excreted at lower titres in pigs, for example O Lausanne (10^6.4^ TCID_50_), O UKG 2001 (10^6.1^ TCID_50_) and South Korea 2000 (10^5.8^ TCID_50_) [23,33]. However, it is important to note that serotype C has not been detected since 2004 and is now considered extinct worldwide [34].

The studies used to measure emissions often differ from one another in regard to the experimental design and as such are not always directly comparable (Figure 2). The most common experimental design used for emission studies is a direct donor-contact challenge design with the donor animal challenged using a needle injection of FMDV into either the tongue, heel bulb or intramuscularly and subsequent emissions measured using a high-volume cyclone sampler (Table 1). Using this experimental set-up Donaldson et al. [35] showed emissions from pigs were between two to three log_10_ TCID_50_ higher than cattle, with the development of generalised lesions in all pigs and cattle, but not for all the sheep, despite comparable virus titres being emitted from cattle and sheep. Alexandersen and Donaldson [23] used a similar experimental design to determine the emission titres of serotype O virus strains, but rather than using a direct contact challenge, they connected the donor and recipient animals through an exposure tunnel. The breath of the donor animal was transported through the tunnel to a fitted mask worn by the recipient animal and emissions from both the donor and recipient pigs were measured using a May three-stage impactor sampler (Table 1).

Together with quantifying emissions from infected animals it is also important to understand the daily emission profiles generated by infected animals to improve estimations of the risk of airborne transmission events. Donaldson [9] reported that all species (cattle, pigs, sheep) excreted virus for four days, with maximal excretion occurring when primary lesions were present (sheep) and at an early stage of generalization (cattle and swine). Furthermore, Gloster et al. [36] showed pigs emitted the highest amount of virus at two days post infection (dpi) when challenged with high (10^5^ TCID_50_) and low (10^3^ TCID_50_) doses of virus (C Noville and O UKG 2001). Emissions increased exponentially and then fell after three dpi and were undetectable after day 4 in the high dose group and day 5 in the low dose group [36]. Alexandersen et al. [21] quantified FMDV in the breath of infected pigs and cattle that had either been needle-inoculated (donors) or had been in direct contact with the donor animals. The authors showed that virus was detectable in the breath of the donor cattle and pigs less than one day after challenge. It took longer for the virus to be detected in the breath of the contact animals, with virus not detectable in the breath until one- and three-days post challenge in cattle and pigs, respectively. The highest emissions from the donor animals were 10^1^ TCID_50_ and 10^2^ TCID_50_ at one day post infection for cattle and pigs respectively. The titre decreased after one day in the breath of the cattle but remained consistent until day four in the breath of the pigs.

### 4.3. Stability of Strains in Aerosols

As well as the amount of virus emitted from infected animals, the stability of such strains in aerosolised FMDV is an important factor in assessing the risk of airborne spread. The stability of virus strains under experimental conditions has previously been investigated using a Collison nebuliser to generate aerosols into a Goldberg drum with the sampling of aerosols using an all-glass Porton impinger sampler (Table 1). Using this method Donaldson [47] compared eight strains of FMDV representing serotypes O, A and C, and found serotype A viruses to be more stable in aerosols compared to the strains from serotypes O and C when exposed to 55% and 70% relative humidity. Donaldson [35] also showed that survival of the virus in aerosols decreased as the humidity level decreased. Similarly, Barlow [39] showed a serotype O virus (O_1_ BFS 1860) had a better survival rate at relative humidity levels above 50% when using the same methodology. Barlow [39] also showed the survival of the virus decreased when aerosols had been stored in the drum for 5 min compared to being sampled immediately after nebulisation. However, it has since been reported that when using rotating drums sampling should not take place immediately but after a mixing period to account for deposition [48,49]. Barlow and Donaldson [40] demonstrated the O_1_ BFS 1860 strain was more stable in aerosols when suspended in bovine saliva than in cell culture media, particularly when the relative humidity level was above 50%. More recently, Brown et al. [46] showed recovery of FMDV in aerosols was slightly higher from the serotype A virus strain tested (A TAI 2016) compared to the strains from serotype O (O UKG 2001) and Asia 1 (Asia1 Shamir 2011), when using a Coriolis micro sampler (Table 1) to collect nebulised aerosols.

In addition to virus aerosols from the breath of infected animals, aerosols can also be generated from other sources such as skin and fomites (bedding, the floor or walls of a pen) which can be re-aerosolised through daily husbandry practices [50]. Gailiunas and Cottral [51] found skin could be a source of aerosolised virus with skin scrapings from cattle containing virus titres of 10^1.0^–10^4.6^ log_10_ PFU. In an experimental setting FMDV has also been detected in aerosols in rooms which had previously been contaminated by infected cattle [6]. This demonstrates FMDV can be resuspended into aerosols from a contaminated environment. Colenutt et al. [6,7] showed that environments surrounding infected animals become contaminated in the field and therefore could provide the opportunity for aerosolisation of virus during movement of animals and people or daily cleaning tasks.

## 5. Geographical Areas Most Likely to Be Affected by Airborne Transmission

The risk of airborne transmission will vary between geographical areas depending on factors such as species present, local climate, animal density and management systems. In many FMD-free countries the climate is generally more temperate with conditions reaching the optimal levels of humidity, temperature and wind speed required to facilitate airborne transmission over long distances [52]. In addition, pigs have been identified as the biggest emitters of airborne virus, so the risk of airborne transmission of FMDV will also depend on the density of pigs in a geographical area. As an example, there is a relatively low density of pigs in sub-Saharan Africa compared to European countries and, as such, the risk of pigs producing an airborne virus plume that would remain strong enough to cause onward transmission of the disease is reduced [53]. Using a Lagrangian stochastic model, Klausner et al. [18] predicted approximately 1000 pigs/km^2^ were required for successful airborne transmission and regions such as the Middle East, even though the scale of pig farming is much smaller, were still at risk of airborne spread. Klausner et al. [18] projected that in Egypt pig populations in certain areas would be sufficient to allow airborne transmission of FMDV and identified countries such as The Netherlands, Denmark and China as being at high risk due to their large pig farming industries.

Many studies have investigated the risk of airborne transmission of FMD into free areas using modelling approaches and as such have identified risk factors specific to airborne transmission. Garner et al. [54] investigated the risk of airborne transmission of FMD during simulated outbreaks in Australia. The results from the model showed that out of 139 farms containing susceptible livestock which were exposed to wind-borne virus, ten (7.2%) premises were identified as medium or high risk, with those closest to the infected premises (IP) at highest risk [54]. There were, however, a few instances where premises further out from the IP were deemed more of a risk, for example a large piggery 40 km away from the IP was determined as medium risk whereas two large beef and sheep farms which were located 27 km from the IP were considered high risk [54]. These premises were identified as being more of a risk because of the high density of animals at the two sites. Garner et al. [54] also identified the winter months, particularly July, as posing the highest risk for airborne spread of FMD. In addition to the presence of large aggregations of intensively farmed animals, the authors stated that in the context of Australia high risk of airborne transmission mostly referred to piggeries and to a lesser extent beef feedlots and dairy farms [54]. In similar studies to assess the risk of airborne transmission across the United States of America, Hagerman et al. [55] showed conditions were most favourable for transport of wind-borne virus during the winter months and in areas with high pig and cattle densities, such as the upper midwestern states. Coffman et al. [56] also found similar results and predicted airborne transmission could cause 36%, 12% and 2% of a 1000 animal herd to become infected when the IP was located 10 km, 15 km and 20 km away respectively. Modelling studies are important for identifying risk factors of airborne transmission of FMD and as such for disease preparedness. The most important risk factors highlighted in these studies were farming intensity, specific weather conditions, distance from the source of infection, and species of livestock.

By comparison, the risk of airborne transmission of FMD in hot climates is relatively understudied, for example in areas of Africa and Asia where the disease is endemic. Instead, modelling has been more commonly used to assess the risk of disease incursions into free countries. There are publications which have modelled airborne transmission of FMD in hot climates but they have used temperate climate data which is not representative of the study area [57]. It is likely that airborne spread of FMD in endemic countries is underrepresented in literature because it is deemed less of a risk than in free countries due the climate being less favourable for airborne spread. It is known the ideal conditions for airborne spread are high humidity levels, low temperatures and a continuous gentle wind. In endemic regions where the climate is hot and dry and the air still, airborne transmission is less likely to occur [58]. However, the use of models to make estimates on the risk of airborne transmission of FMD using data specific to endemic regions, for example, climate information and epidemiological data on circulating strains and their behaviour in aerosols, would still be of value in assessing the level of risk of airborne spread in these regions which could inform control measures.

## 6. Instrumentation Used to Detect FMDV

Many different types of instrumentation have been used to measure FMDV in aerosols. The advantages and disadvantages of the samplers most commonly used for detection of FMDV are outlined in Table 1.

To date, there has not been a sampler specifically designed for use with FMDV and most samplers used in FMDV studies were developed in the 1950s and 1960s. Many of the samplers were designed originally for the field of air quality and although samplers have since been developed for use with bioaerosols few have been tested with FMDV. Ideally, an air sampler suitable for collecting FMDV aerosols would be robust, lightweight, and compact for transportation. The sampler should also be easy to operate and to clean, without damaging sensitive parts by repeat disinfection. It is also preferable when sampling in close proximity to animals that the sampler is quiet when in operation. For use in field work, an aerosol sampler needs to be capable of operating and being stored in hot climates and have a long battery life for sampling in remote locations. Often the samplers used in the original FMDV studies were made of glass (SKC BioSampler, May and Porton), making them easy to clean and disinfect but at the expense of robustness and portability [29,37,41]. Some of the samplers also required the use of an external pump (SKC BioSampler and May) adding to the weight and size of the instrument as well as being loud when in operation [41].

A large range of different types of samplers and their operation have been reported in the literature, but the most popular for use in FMDV studies are cyclones and liquid impingers. These samplers have the benefit of an increased chance of virus survival which is facilitated by elution into liquid [59]. Doel et al. [38] compared the efficiency of four air samplers: Cyclone, Porton, May and the SKC BioSampler samplers to detect FMDV under laboratory conditions. The Porton and SKC BioSampler samplers are liquid impingers which work using a pump to draw in surrounding air into liquid [37]. The Cyclone draws in air which is impacted on the sides of a collection vessel and particles are then washed into liquid [38]. The May is a three-stage impactor which consists of three collection plates that reduce in size to determine the size of particle sampled [41]. Doel et al. [38] found that overall the Cyclone performed slightly better than the other three samplers tested as infectious virus was detected in samples collected on three sampling days, as opposed to only two days with the other samplers tested. However, it was concluded that no single sampler was optimal for every situation, and the use of a combination of samplers was recommended [38]. Nelson et al. [42] also showed that the Cyclone was most sensitive at detecting FMDV at different stages of disease (preclinical, clinical and recovery) when compared to the May, SKC BioSampler, BioBadge and the Airport MD8 samplers.

More recently, there has been more options available for samplers specifically developed for bioaerosol research and diagnostic use, with a move towards portability, robustness and ease of use than previous samplers. The Coriolis micro sampler has recently been successfully employed to detect FMDV when in close proximity to infected animals both under laboratory conditions and in field settings [6,7]. Recent outbreaks of respiratory viruses causing serious threats to public health such as influenza H1N1 and the ongoing COVID-19 pandemic have brought into focus the importance of studying viruses in aerosols [60]. Aerosol sampling in hospitals and at live bird markets have resulted in detection of SARS-CoV-2 and avian influenza respectively [61,62,63,64]. However, despite a general increase in these types of studies there are still challenges for detecting viral pathogens in air samples, including sampling methods negatively affecting viral infectivity and availability of standardised protocols for assessing both the physical and biological efficiency of samplers [59,60].

## 7. The Use of Mathematical Models to Quantify and Predict Airborne Transmission

Simulating the airborne spread of FMD can be accomplished by using an atmospheric dispersion model. If implemented alongside other surveillance tools and methods, dispersion calculations can be used tactically to aid the decision-making processes directly following an outbreak [65,66,67]. In addition, together with other multi-level transmission models, dispersion models may be used to examine the likelihood of disease spread given the configuration of farms nationally. The results can be used as basic data in the planning of the national preparedness for FMD outbreaks [54,56,65,68]. These computer models use information about the emitted virus aerosols and the weather to describe the movement of the virus plume in the atmosphere, indicating where it will travel and what concentrations are likely at downwind locations [69]. Consequently, these models can be used to estimate the risk of airborne transmission of FMD and if used with forecast weather data, can possibly indicate likely downwind concentrations a day or two in advance [70].

Essential elements for the operation of a dispersion model are input data streams which can be collected over a suitable period of time from either direct measurements or inferences from empirical formula. More specifically, the input data needed include information on:(i)Meteorological conditions: wind direction and speed (which will indicate where the plume will travel and how fast) [71]. Data including mixing height and ambient temperature will allow the model to determine how stable the atmosphere is and therefore how conducive to turbulent motion which will help dilute the released virus aerosol—leading to lower downwind virus concentrations [72]. Relative humidity levels are important as values below 55% will impact on the viability of the virus in transit.(ii)Some models will be able to accommodate the impact of complex terrain and the presence of orographic features (e.g., mountains, steep-sided hills, valleys or coast) at the source premise(s), pathways, and at the receptor sites, as these features may affect the general flow patterns impacting the virus plume [71]. More advanced dispersion models can make use of full three-dimensional meteorological data to describe the general atmospheric movement. The UK Met Office uses the Numerical Atmospheric Modelling Environment (NAME) [73] as its model of choice to describe airborne FMDV aerosol and benefits from receiving three-dimensional meteorological fields from the Met Office’s Numerical Weather Prediction model, the Unified Model [74].(iii)Virus emission rates: the concentration of virus being emitted into the environment and the duration of virus shedding [71]. Virus emission time series are a crucial element informing the model of the atmospheric loading due to the virus aerosols. This will inform the model about the titre of virus emitted to the atmosphere and how this might vary with time [54]. As FMDV is susceptible to certain environmental conditions, dispersion models can be modified to allow for removal and deterioration of the virus due to external environmental factors (such as wet deposition due to rainfall or changes in humidity) [54].

Although dispersion models can be very useful, much will depend on the quality of the input data and the sophistication of the inherent science incorporated in the model. Some models will describe the general atmospheric motion well and others not so well. Gloster et al. [75] reported on a model intercomparison for the airborne spread of the disease organised by The Pirbright Institute (then the Institute for Animal Health), and the UK Met Office. The models involved attempted to describe the airborne spread relating to the UK outbreak of FMD in 1967. All the models identified similar livestock premises at risk, with any main differences put down to differences in the meteorology used for the analysis. Another key finding highlighted the importance of an accurate description of the sequence of events occurring on the infected farms—timing and duration of virus emissions from all infected animals. This was particularly true for cases where the prevailing meteorology varied substantially during the virus emission period. In more general terms, differences in the assumptions concerning the virus release, environmental fate and the susceptibility of naïve livestock to airborne infection in downwind locations, can all impact substantially on the areas deemed to be at risk by the dispersion models.

## 8. Future Research

Most of the previous studies centred on FMDV in aerosols have focused on quantifying emissions from infected animals and the survival of virus in aerosols. However, they have yet to fully characterise and quantify virus emissions from infected animals and contaminated environments, which provides a rich area for further study. Dust particles are generated constantly at varying levels within livestock farms and certain procedures (e.g., movement of animals, pressurised water cleaning or the action of farm vehicles) may lead to the resuspension of dust containing virus from surfaces. Aerosols from infected farms may carry varying levels of virus which will depend on the characteristics (particle size and composition) of the dust particles generated within a farm. A clearer understanding of how these processes generate aerosols would be useful to provide information on the distance of detection, as well as providing better predictions of farm-level emissions and, hence, the risk of airborne spread. Furthermore, it would be useful for modelling approaches to ascertain if changes in the prevailing weather (for example temperature and humidity) influence particle size, or if there are differences in emission profiles between healthy and infected animals. Therefore, in the future research instruments such as particle size spectrometers or wireless-sensor networks could be utilised to characterise the particle size of aerosolised dust from different farming systems. The application of such equipment could also be extended to the study of other livestock diseases that are prevalent and have large impact on farming economics, such as Porcine Reproductive and Respiratory Syndrome virus (PRRSV) or Avian and Swine influenza (SwIV and AIV).

The survival of FMDV in aerosols and the difference in decay rates between serotypes and strains of FMDV has been investigated, but most of these studies were conducted in the 1980s using strains which are no longer circulating. There is a gap in knowledge regarding the survival rate of current, circulating strains and the trade–off, if any, between virulence and survival. Laboratory experiments to investigate the survival of FMDV strains when exposed to different environmental conditions, such as varying levels of humidity and temperature, could help bridge this gap in knowledge. Several approaches exist to suspend infectious viral pathogens in aerosols and expose them to different environmental conditions, such as using a Goldberg drum to hold aerosols in suspension, the microthreads technique designed to capture and suspend aerosols on spider webs and the capture and levitation of single or multiple bioaerosol droplets using the controlled electrodynamic levitation and extraction of bioaerosol onto a substrate (CELEBS) technique [76,77,78,79]. If similar data were to be generated for contemporary FMDV strains it could be utilised to improve the estimates of airborne transmission risks for different strains and across different geographical regions calculated by dispersion models.

Another important area that is relatively understudied in the field of FMD research is the use of air sampling as a surveillance tool in FMD control programmes. Colenutt et al. [7] showed that FMDV could be detected in aerosol samples taken on small holdings, in close proximity to infected cattle. This work could be expanded further to better understand the use of aerosol sampling at the herd-level, both in endemic and outbreak settings. A sampler which can be run continuously, requires little manual operation and is easy to use would be required for this type of research. However, the Coriolis micro sampler used previously in field settings has the problem of evaporation of collection medium over time (Table 1), making it unsuitable in its current form for continuous measurements over a long sampling period. The ideal sampler for this type of work would need to be portable, robust and easy to clean for the transportation and disinfection procedures required. Future research in this vein may require field studies that seek to trial existing samplers, or development with engineering partnerships to produce a suitable candidate sampler for this work.

## 9. Conclusions

Transmission of FMDV via aerosols occurs less frequently than direct contact transmission. However, the importance of this indirect route of transmission has been demonstrated not only through experimental studies which aim to quantify emissions from infected animals but also in real-life outbreak situations. Both epidemiological studies and mathematical modeling have provided strong evidence that spread of disease over short and long distance was due to FMDV transported in aerosols. There is a large body of research on emissions from animals and how this varies over time and depends on host species and virus strain. This data has been useful in improving our understanding of the behaviour of FMDV in aerosols and the transmission risk they pose. However, the parameters often differ from study to study which can make it difficult to draw firm conclusions from the data and the data does not encompass current circulating strains of FMDV. Most of the published literature describes outdated methodology with samplers which are no longer available, while those currently available have not been designed for working with pathogens in high containment facilities or in rural, field locations. Further information regarding the survival of different FMDV strains and emission data from farms housing susceptible animals is needed to provide better and more up-to-date input data into mathematical models to make more accurate predictions of the risk of airborne spread.

## Figures and Tables

**Figure 1 viruses-14-01009-f001:**
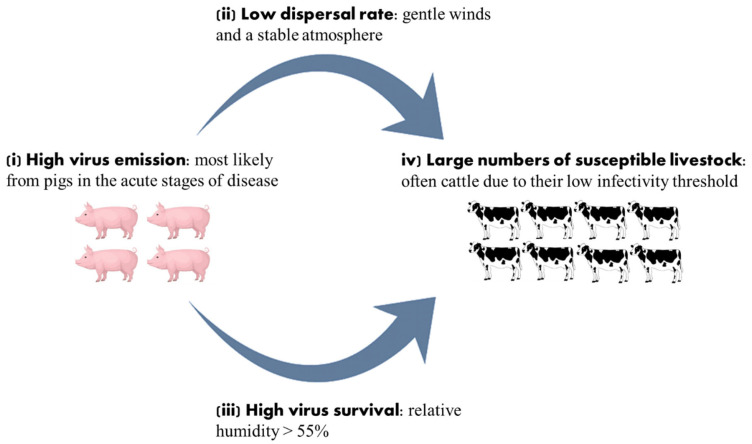
The factors required for airborne transport of foot-and-mouth disease virus aerosols over long distances.

**Figure 2 viruses-14-01009-f002:**
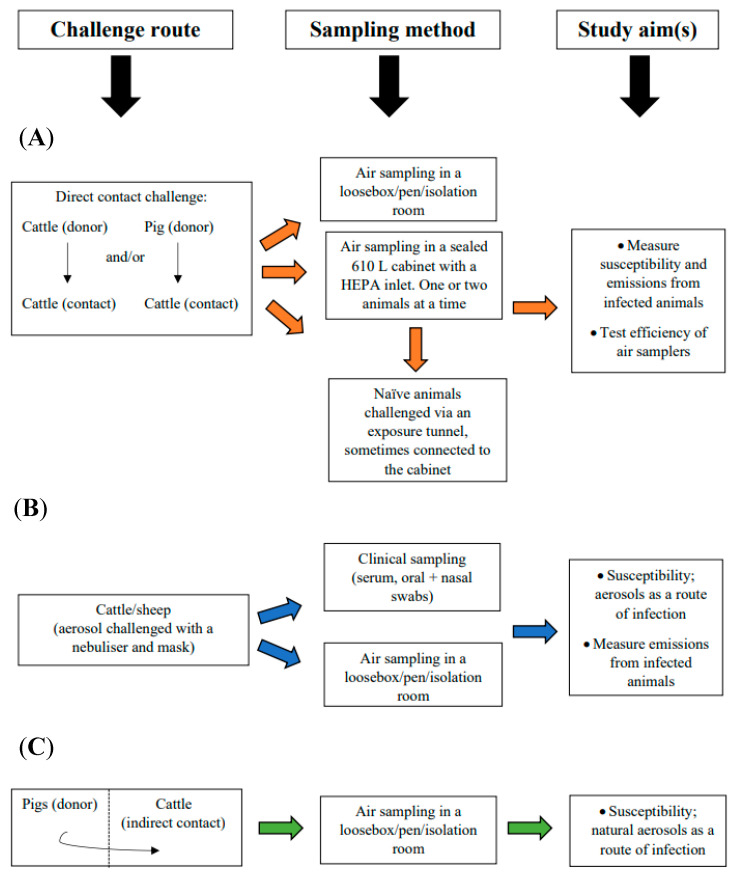
The three most commonly used experimental designs for studying FMDV in aerosols. (**A**) A direct contact challenge between a needle-inoculated donor and a contact animal, with subsequent air sampling either within a loosebox or a sampling cabinet (orange arrows). Some designs connected an exposure tunnel to the sampling cabinet for an indirect challenge to measure susceptibility and emissions. (**B**) Aerosol challenge experiments using a nebuliser and mask to deliver aerosols, followed by air and clinical sampling to assess virus shedding (blue arrows). (**C**) A between-pen challenge design using needle-inoculated donors and indirect contact recipient animals in separate rooms with follow up sampling to assess viral shedding (green arrows).

**Table 1 viruses-14-01009-t001:** Examples of air samplers previously used to measure FMDV in aerosols.

Instrument	Method of Operation	Size Range	Sampling Mode, Volume, and Time	Availability	Advantages	Disadvantages	Examples of Use
Porton	Liquid impinger	<18 µm	On demand, 11 L/min for 5 min	Obsolete	Easy to disinfect Elution into medium	Fragile	[37,38,39,40]
May	Three stage impactor	1st stage- >6 µm 2nd stage- 3–6 µm 3rd stage 3–0.8 µm	On demand, 33 L/min for 5 min	Obsolete	Allows for particle separation Elution into medium	Requires an external pump Fragile	[38,41,42]
Cyclone	Particles impacted on the sides and washed by impinger fluid	<18 µm	On demand, 390 L/min for 10 min	Obsolete	High flow rate Elution into medium	Requires an external pump	[38,42,43]
BioBadge	Particles are driven onto a disposable rotating disc	3–8 µm	Continuous, 10 L/min for 3–20 h	Obsolete	Long run time Small and light Little manual operation	Low sampling efficiency	[29,42,44]
SKC BioSampler	Liquid impinger	<18µm	On demand, 12.5 L/min for 15 min	Commercial	Elution into medium	Low flow rate	[29,38,42]
BioCapture 650	Rotating impeller arms drive particles against a plastic wall and washed by collecting fluid	0.5–10 µm	On demand, 200 L/in for 30 min	Commercial	Handheld Robust Little manual operation	Short battery life Heavy	[44]
Airport MD8	Gelatine membrane filter	0.65–3 µm	On demand, 50 L/min for 10 min	Commercial	Handheld Quiet operation	pH of filters not compatible with isolation of live FMDV	[29,42]
Dry Filter Unit, model 1000	Polyester felt filter	>1 µm	Continuous, 144 L/min (pump dependant) for 12 h	Commercial	Long run time Little manual operation	Requires mains power Not tested live virus recovery	[31,45]
Coriolis micro	Particles impacted on the sides and washed by impinger fluid	>0.5 µm	On demand, 300 L/min for 10 min	Commercial	High flow rate. Elution into medium	Can corrode after disinfection Evaporation of collection medium	[6,7,46]

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
