# Peer review of "Airborne Transmission of Foot-and-Mouth Disease Virus: A Review of Past and Present Perspectives"

_viruses, 2022, doi:10.3390/v14051009_

Round 1
Reviewer 1 Report
This review specifically describes and discusses the transmission of FMDV in aerosols, by focusing on the nature, the generation, transmission distance, susceptible animal species, instruments and methods for the investigation. The review revisits an adequate number of references to support its conclusion. Figures in the review can help facilitate to understand the paper. Generally, this review is qualified for publication after minor modification. The modifications are as follows, but not limited to these:
----in Figure 1, low dispersal rate and high virus survival should be explained clearly.
---- In Figure 2, A: should point out the challenge route.
----Between 144-147, 108.6 TCID50 and 107.6 TCID50.......: here 8.6 and 7.6 should be superscript.
---- A needle-inoculated animals should be the same in the whole text.
---- In line 92, “60Km to 150 Km” should be “60 to 150 Km”. The expression should be consistent all the time.
Author Response
Thanks to the reviewers for helpful comments on clarifying and improving some points of the manuscript. Please see below for how these comments have been addressed.
Reviewer one:
Point 1: in Figure 1, low dispersal rate and high virus survival should be explained clearly.
Response 1: Figure 1 has been edited to include more details on each of the factors required for airborne transport of foot-and-mouth disease virus
Point 2: In Figure 2, A: should point out the challenge route.
Response 2: Figure 2 and it’s legend have been edited to clarify that experimental design A is a direct contact challenge.
Point 3: Between 144-147, 108.6 TCID50 and 107.6 TCID50.......: here 8.6 and 7.6 should be superscript.
Response 3: I believe this must have been a formatting issue with the upload of the manuscript or potentially an incompatibility between word versions. I have uploaded the revised manuscript as a PDF (as well as a word document) which will hopefully display the symbols correctly.
Point 4: Between A needle-inoculated animals should be the same in the whole text.
Response 4: This has been changed throughout the text to ‘needle-inoculated’.
Point 5: In line 92, “60Km to 150 Km” should be “60 to 150 Km”. The expression should be consistent all the time.
Response 5: This line has been changed within the text.
Reviewer 2 Report
The manuscript is a review of the literature on the airborne transmission of foot-and-mouth disease among livestock. The review is thorough and well-written. I have only a few minor comments.
In the Introduction on page 2, the authors say:
“It is generally accepted that droplet transmission is a form of direct contact transmission in which respiratory droplets measuring >5 µm in diameter travel directly from the respiratory tract of an infectious host to a susceptible host, over short distances (<1 metre) [13,14]. By contrast, airborne transmission is defined as respirable particles which are exhaled from an infectious host and partially evaporate in the surrounding air [15]. These small, partially evaporated particles measuring <5 µm in diameter can remain infectious over time and can be dispersed over long distances (>1 metre) by air currents, potentially causing long distance transmission events [13,14].”
The authors are absolutely correct that these are the conventional definitions of airborne and droplet transmission that are widely used. However, these definitions are not based in aerosol science, and in reality particles much larger than 5 µm can remain airborne for a significant time and be inhaled. Unfortunately, during the current COVID-19 pandemic, these definitions were used to justify the erroneous assumption that COVID-19 was not spread by aerosols. Several prominent aerosol researchers have been urging that the term aerosol transmission be used for particles less than 100 µm, and that droplet transmission be used only for larger particles that truly do settle rapidly.
I recognize that the authors may not wish to get into this issue, and if they choose to leave the current definitions, then I would certainly accept that decision. However, I would encourage them to read through some of the recent discussion of this topic and consider rewriting this section:
Jimenez, J, L Marr, K Randall, ET Ewing, Z Tufekci, T Greenhalgh, DK Milton, R Tellier, J Tang, Y Li, L Morawska, J Mesiano-Crookston, D Fisman, O Hegarty, S Dancer, P Bluyssen, G Buonanno, M Loomans, W Bahnfleth, M Yao, C Sekhar, P Wargocki, AK Melikov and K Prather (2021). Echoes Through Time: The Historical Origins of the Droplet Dogma and its Role in the Misidentification of Airborne Respiratory Infection Transmission. SSRN Electronic Journal (preprint). https://doi.org/10.2139/ssrn.3904176
Marr, LC and JW Tang (2021). A Paradigm Shift to Align Transmission Routes With Mechanisms. Clin Infect Dis 73(10): 1747-1749. https://doi.org/10.1093/cid/ciab722
Prather, KA, LC Marr, RT Schooley, MA McDiarmid, ME Wilson and DK Milton (2020). Airborne transmission of SARS-CoV-2. Science 370(6514): 303-304. https://doi.org/10.1126/science.abf0521
Tang, JW, WP Bahnfleth, PM Bluyssen, G Buonanno, JL Jimenez, J Kurnitski, Y Li, S Miller, C Sekhar, L Morawska, LC Marr, AK Melikov, WW Nazaroff, PV Nielsen, R Tellier, P Wargocki and SJ Dancer (2021). Dismantling myths on the airborne transmission of severe acute respiratory syndrome coronavirus-2 (SARS-CoV-2). J Hosp Infect 110: 89-96. https://doi.org/10.1016/j.jhin.2020.12.022
I would also note that the authors are correct that the aerosol particles of concern in the transmission of FMDV over long distances are less than 5 µm since larger particles would almost certainly settle before they could travel a kilometer or more.
I noticed a few formatting problems in the paper:
In the section cited above, the micro symbol is replaced by a question mark.
The numbers for the TCID50 are incorrectly formatted in several places in that the exponent part of the number is not superscripted. For example, the text is formatted as “The highest emissions recorded for C Noville were 108.6 144 TCID50 and 107.6 TCID50 in pigs [19,30]” rather than “The highest emissions recorded for C Noville were 10 (superscript 8.6) TCID50 and 10 (superscript 7.6) TCID50 in pigs [19,30].”
Finally, the authors say:
“For example, existing samplers are not necessarily compatible with the logistical challenges of working in a high containment facility or in remote and/or rural locations for fieldwork. Ideally, an air sampler suitable for collecting FMDV aerosols would be robust, lightweight, and compact for transportation. The sampler should also be easy to operate and to clean, without damaging sensitive parts by repeat disinfection.”
Several high-flowrate wetted wall cyclone and filter-based samplers have been developed for biodefense and biosecurity applications, although I don’t know if they have been tried for FMDV. The authors mention the BioBadge and BioCapture, but others are available as well that they may find to be more suitable. The NIOSH Manual of Analytical Methods has a list of these in the chapter on bioaerosol sampling (https://www.cdc.gov/niosh/nmam/pdf/chapter-ba.pdf, Tables I-VI).
Author Response
Thanks to the reviewers for helpful comments on clarifying and improving some points of the manuscript. Please see below for how these comments have been addressed.
Reviewer two:
Point 1: In the Introduction on page 2, the authors say:
“It is generally accepted that droplet transmission is a form of direct contact transmission in which respiratory droplets measuring >5 µm in diameter travel directly from the respiratory tract of an infectious host to a susceptible host, over short distances (<1 metre) [13,14]. By contrast, airborne transmission is defined as respirable particles which are exhaled from an infectious host and partially evaporate in the surrounding air [15]. These small, partially evaporated particles measuring <5 µm in diameter can remain infectious over time and can be dispersed over long distances (>1 metre) by air currents, potentially causing long distance transmission events [13,14].”
The authors are absolutely correct that these are the conventional definitions of airborne and droplet transmission that are widely used. However, these definitions are not based in aerosol science, and in reality particles much larger than 5 µm can remain airborne for a significant time and be inhaled. Unfortunately, during the current COVID-19 pandemic, these definitions were used to justify the erroneous assumption that COVID-19 was not spread by aerosols. Several prominent aerosol researchers have been urging that the term aerosol transmission be used for particles less than 100 µm, and that droplet transmission be used only for larger particles that truly do settle rapidly.
I recognize that the authors may not wish to get into this issue, and if they choose to leave the current definitions, then I would certainly accept that decision. However, I would encourage them to read through some of the recent discussion of this topic and consider rewriting this section:
Jimenez, J, L Marr, K Randall, ET Ewing, Z Tufekci, T Greenhalgh, DK Milton, R Tellier, J Tang, Y Li, L Morawska, J Mesiano-Crookston, D Fisman, O Hegarty, S Dancer, P Bluyssen, G Buonanno, M Loomans, W Bahnfleth, M Yao, C Sekhar, P Wargocki, AK Melikov and K Prather (2021). Echoes Through Time: The Historical Origins of the Droplet Dogma and its Role in the Misidentification of Airborne Respiratory Infection Transmission. SSRN Electronic Journal (preprint). https://doi.org/10.2139/ssrn.3904176
Marr, LC and JW Tang (2021). A Paradigm Shift to Align Transmission Routes With Mechanisms. Clin Infect Dis 73(10): 1747-1749. https://doi.org/10.1093/cid/ciab722
Prather, KA, LC Marr, RT Schooley, MA McDiarmid, ME Wilson and DK Milton (2020). Airborne transmission of SARS-CoV-2. Science 370(6514): 303-304. https://doi.org/10.1126/science.abf0521
Tang, JW, WP Bahnfleth, PM Bluyssen, G Buonanno, JL Jimenez, J Kurnitski, Y Li, S Miller, C Sekhar, L Morawska, LC Marr, AK Melikov, WW Nazaroff, PV Nielsen, R Tellier, P Wargocki and SJ Dancer (2021). Dismantling myths on the airborne transmission of severe acute respiratory syndrome coronavirus-2 (SARS-CoV-2). J Hosp Infect 110: 89-96. https://doi.org/10.1016/j.jhin.2020.12.022
I would also note that the authors are correct that the aerosol particles of concern in the transmission of FMDV over long distances are less than 5 µm since larger particles would almost certainly settle before they could travel a kilometer or more.
Response 1: A few sentences have been added to the aerosol vs droplet paragraph to acknowledge that the current definitions have been disputed due to new evidence from COVID-19 studies. But given the distances FMDV can travel in aerosols we still define aerosol transmission as particles measuring <5 μm in diameter.
Point 2: In the section cited above, the micro symbol is replaced by a question mark.
Response 2: I believe this must have been a formatting issue with the upload of the manuscript or potentially an incompatibility between word versions. I have uploaded the revised manuscript as a PDF (as well as a word document) which will hopefully display the symbols correctly.
Point 3: The numbers for the TCID50 are incorrectly formatted in several places in that the exponent part of the number is not superscripted. For example, the text is formatted as “The highest emissions recorded for C Noville were 108.6 144 TCID50 and 107.6 TCID50 in pigs [19,30]” rather than “The highest emissions recorded for C Noville were 10 (superscript 8.6) TCID50 and 10 (superscript 7.6) TCID50 in pigs [19,30].”
Response 3: Again, I believe this must have been a formatting issue with the upload of the manuscript or potentially an incompatibility between word versions. Hopefully uploading a PDF version of the revised manuscript will solve this.
Point 4: Finally, the authors say:
“For example, existing samplers are not necessarily compatible with the logistical challenges of working in a high containment facility or in remote and/or rural locations for fieldwork. Ideally, an air sampler suitable for collecting FMDV aerosols would be robust, lightweight, and compact for transportation. The sampler should also be easy to operate and to clean, without damaging sensitive parts by repeat disinfection.”
Several high-flowrate wetted wall cyclone and filter-based samplers have been developed for biodefense and biosecurity applications, although I don’t know if they have been tried for FMDV. The authors mention the BioBadge and BioCapture, but others are available as well that they may find to be more suitable. The NIOSH Manual of Analytical Methods has a list of these in the chapter on bioaerosol sampling (https://www.cdc.gov/niosh/nmam/pdf/chapter-ba.pdf, Tables I-VI).
Response 4: The first sentence cited above has been removed and the previous sentence has been expanded to say that there are samplers used with bioaerosols but few of these samplers have been trialled with FMDV.